palaeontology

palaeopathology, palaeontology, ichthyosauria, palaeoecology, Posidonienschiefer Formation, marine reptiles

**Author for correspondence:**
Judith M. Pardo-Pérez
e-mail: judith.pardo-perez@smns-bw.de

# Palaeoepidemiology in extinct vertebrate populations: factors influencing skeletal health in Jurassic marine reptiles

Judith M. Pardo-Pérez[1,2], Benjamin Kear[3] and Erin E. Maxwell[1]

[1]Staatliches Museum für Naturkunde, Stuttgart, Germany
[2]Dirección de Investigación y Postgrado, Universidad de Magallanes, Punta Arenas, Chile
[3]Museum of Evolution, Uppsala University, Uppsala, Sweden

 JMP-P, 0000-0001-9477-1110; BK, 0000-0002-3128-3141; EEM, 0000-0002-6032-6251

Palaeoepidemiological studies related to palaeoecology are rare, but have the potential to provide information regarding ecosystem-level characteristics by measuring individual health. In order to assess factors underlying the prevalence of pathologies in large marine vertebrates, we surveyed ichthyosaurs (Mesozoic marine reptiles) from the Posidonienschiefer Formation (Early Jurassic: Toarcian) of southwestern Germany. This Formation provides a relatively large sample from a geologically and geographically restricted interval, making it ideal for generating baseline data for a palaeoepidemiological survey. We examined the influence of taxon, anatomical region, body size, ontogeny and environmental change, as represented by the early Toarcian Oceanic Anoxic Event, on the prevalence of pathologies, based on *a priori* ideas of factors influencing population skeletal health. Our results show that the incidence of pathologies is dependent on taxon, with the small-bodied genus *Stenopterygius* exhibiting fewer skeletal pathologies than other genera. Within *Stenopterygius*, we detected more pathologies in large adults than in smaller size classes. Stratigraphic horizon, a proxy for palaeoenvironmental change, did not influence the incidence of pathologies in *Stenopterygius*. The quantification of the occurrence of pathologies within taxa and across guilds is critical to constructing more detailed hypotheses regarding changes in the prevalence of skeletal injury and disease through Earth history.

# 1. Introduction

Palaeoepidemiological surveys have the potential to provide novel information on individual health, as well as overarching metrics of ecosystem function such as predation pressure. Such surveys have been attempted in numerous fossil systems (e.g. non-avian dinosaurs [1], and Pleistocene carnivorans [2] and frogs [3]); however, baseline data are largely lacking for ancient marine systems. Such data can provide not only new information relevant to the palaeoecology of specific taxa, but can also shed light on the relationship between the prevalence of skeletal trauma/disease and life history, trophic level and environmental parameters. The possibility of surveying a relatively large sample of individuals from a geologically and geographically restricted interval spanning a range of trophic levels and ontogenetic stages makes ichthyosaurs from southwestern Germany an ideal system to provide baseline data for this type of palaeoepidemiological survey.

Ichthyosaurs are a group of Mesozoic marine reptiles that occupied a broad range of trophic levels in Early Jurassic pelagic ecosystems [4,5]. The ichthyosaurian fossil record is controlled by lagerstätte-type deposits, in which temporally and geographically restricted formations yield large numbers of associated skeletons [6]. An example of such is the Posidonienschiefer Formation, a classic konservat lagerstätte yielding a rich assemblage of exceptionally preserved fossil vertebrates and invertebrates, spanning a minor mass extinction associated with ecosystem-level changes in vertebrate and invertebrate faunas (the early Toarcian Oceanic Anoxic Event (T-OAE) [7,8]). Ichthyosaurs are the most abundant large vertebrates from this formation, and are represented in museum collections worldwide by hundreds of complete, articulated skeletons [9]. Five ichthyosaur genera are known from these deposits, varying in body size and trophic position: *Hauffiopteryx* (2.5 m long), *Stenopterygius* (3.5 m long), *Suevoleviathan* (4 m long), *Eurhinosaurus* (7 m long) and the apex predator *Temnodontosaurus* (greater than 9 m long).

Previous studies on the distribution of pathologies in ichthyosaurs have suggested that the actual prevalence of skeletal trauma is much higher than reflected in the literature; however, the anatomical distribution of pathologies is accurately reflected in published reports [10,11]. These studies were, respectively, based on a meta-analysis of published reports [10] and a detailed survey of pathologies within a single genus [11]. Factors affecting the overall incidence of osteopathologies have not been quantified across taxa within a limited temporal and geographical scope. Here, we consider three categories of pathological bone modifications, following [10]: traumatic injury, articular disease and ankylosis. We examine the effect of body size, ontogeny and the T-OAE on the prevalence and distribution of pathologies in ichthyosaurs from the Early Jurassic Posidonienschiefer Formation of southwestern Germany in order to statistically assess which factors affect observed prevalence. We hypothesize that (i) larger taxa occupying higher trophic levels will exhibit a greater incidence of traumatic injuries caused by intraspecific aggression, (ii) smaller taxa may exhibit frequent incidences of joint disease and avascular necrosis, but overall will have fewer traumatic injuries, and (iii) juveniles will exhibit the fewest pathological skeletal modifications, due to more rapid healing or reduced survivorship following trauma. This approach will provide a critical baseline for measuring changes in the prevalence of osteopathologies through time and across habitats in large marine vertebrates.

## 1.1. Institutional abbreviations

SMNS, Staatliches Museum für Naturkunde Stuttgart, Germany; GPIT, Geologisches und Paleontologisches Institut Tübingen, Germany; UMH, Urwelt Museum Hauff; FWD, Holcim Museum, Dotternhausen, Germany; PMU, Paleontological Museum, Uppsala University, Sweden.

# 2. Material and methods

To assess factors influencing the prevalence of osteopathologies in ichthyosaurs, we surveyed 236 specimens from the Posidonienschiefer Formation housed in the SMNS, GPIT, UMH and FWD collections [12], and scored the presence/absence of pathologies. We divided the skeleton into five anatomical regions following [10,13] to account for variation in skeletal completeness: skull, ribs and gastralia, vertebral column, pectoral girdle and forefin, and pelvic girdle and hind fin. In addition, we classified observed pathologies into three types following [10]: trauma with evidence of healing, articular disease and ankyloses. A single observer (J.M.P.-P.) collected all data in order to control for inter-observer detection bias.

We examined the influence of four variables (genus, size, ontogeny and stratigraphic provenance) on the presence and absence of pathologies. Generic body size was based on the categories used by

McGowan & Motani [4]: small–medium (adults < 4 m long) versus large (adults ≥ 4 m long). For the ontogenetic and stratigraphic analyses, we considered only *Stenopterygius* due to large available sample size. We grouped *Stenopterygius* into four size classes, based on estimated mandibular length: neonates and juveniles (mandibular length estimated at less than 40 cm), small adults (mandibular lengths between 40 and 50 cm), large adults (mandibular lengths between 50 and 60 cm) and very large adults (mandibular lengths greater than 60 cm). None of the specimens in the largest size category are attributed to *Stenopterygius quadriscissus*, as the largest confirmed specimen of this species has a mandibular length of 56.4 cm [14], but includes individuals of both *S. triscissus* and *S. uniter*. For the stratigraphic analysis, we classified specimens into three time intervals, based on the position of the invertebrate extinction horizon in the Southwest German Basin [8,15]: $\varepsilon I_1$–$II_3$ (before the anoxic event), $\varepsilon II_{4-5}$ (beds which represent the most severe and prolonged development of anoxic conditions in the basin [16]) and the upper part of the section ($\varepsilon II_6$–III), representing a gradual return to normal marine conditions characterized by fluctuating benthic oxygenation [16]. For the taxonomic and ontogenetic analyses, we considered all surveyed specimens ($n = 233$) and included anatomical region as a covariate to control for differences in completeness between taxa and size classes (e.g. [17]). However, for the stratigraphic provenance analysis, we considered only specimens preserving greater than 50% (105/233) of the skeleton to eliminate anatomical region as a covariate, thus maximizing statistical power. When anatomical region was used as a covariate, we also tested the interaction term between the variable and the covariate, in order to consider the possibility that the anatomical distribution of pathologies was affected by the primary variable (e.g. in the case of taxon, that different areas of the skeleton were preferentially affected in different genera). If not significant, the analysis was rerun without the interaction term to increase statistical power. We analysed the data using binomial logistic regression, as this test accommodates multiple categorical predictor variables (e.g. taxon + anatomical region) and a binary response variable (pathology: present/absent), in the base package of the software platform R [18].

# 3. Results

## 3.1. Types of pathologies observed

### 3.1.1. Stenopterygius

Seventeen of 171 specimens of *Stenopterygius* were pathological [12]. Trauma with signs of healing was the most frequent type of pathology in *Stenopterygius*, observed in 13 cases, most often affecting the ribs and gastralia (8/13) (figures 1*d* and 2). In three specimens (PMU 24320, UMH/dc10, SMNS 9808), injured ribs are associated with infection and abscesses. PMU 24320 also shows a traumatic injury with abscesses related to an infection in the mandible [10]. Three specimens of *Stenopterygius* show pathologies in the pectoral girdle and forefin involving the ankylosis of limb elements (GPIT no. 83; SMNS 56856; SMNS 50007; figure 1*a*,*b*). Vertebral column pathologies have been observed in only two specimens, both showing ankylosis of the neural spines (GPIT/RE/7301; PMU 24320).

### 3.1.2. Hauffiopteryx

Two of eight examined specimens exhibited pathologies (figure 2) [12]. SMNS 81367 showed a gastralium with a callus developed (figure 1*c*) and broken and healed centra in the posterior vertebral column. SMNS 80226 showed three broken and healed gastralia, with signs of infection in two of them.

### 3.1.3. Eurhinosaurus

Of 23 specimens of *Eurhinosaurus* surveyed, five were pathological (figure 2) [12]. The most affected anatomical units were the ribs (two specimens) and pectoral girdle and forefin (two specimens). Both of the latter specimens were affected by ankylosis between phalanges [11]. No pathologies were observed in the vertebral column or pelvic girdle/hind fin.

### 3.1.4. Temnodontosaurus

Of 27 specimens of *Temnodontosaurus* surveyed, eight were pathological (figure 2) [12]. The most frequently affected anatomical unit is the skull, with bone fibre remodelling in the jaws observed in

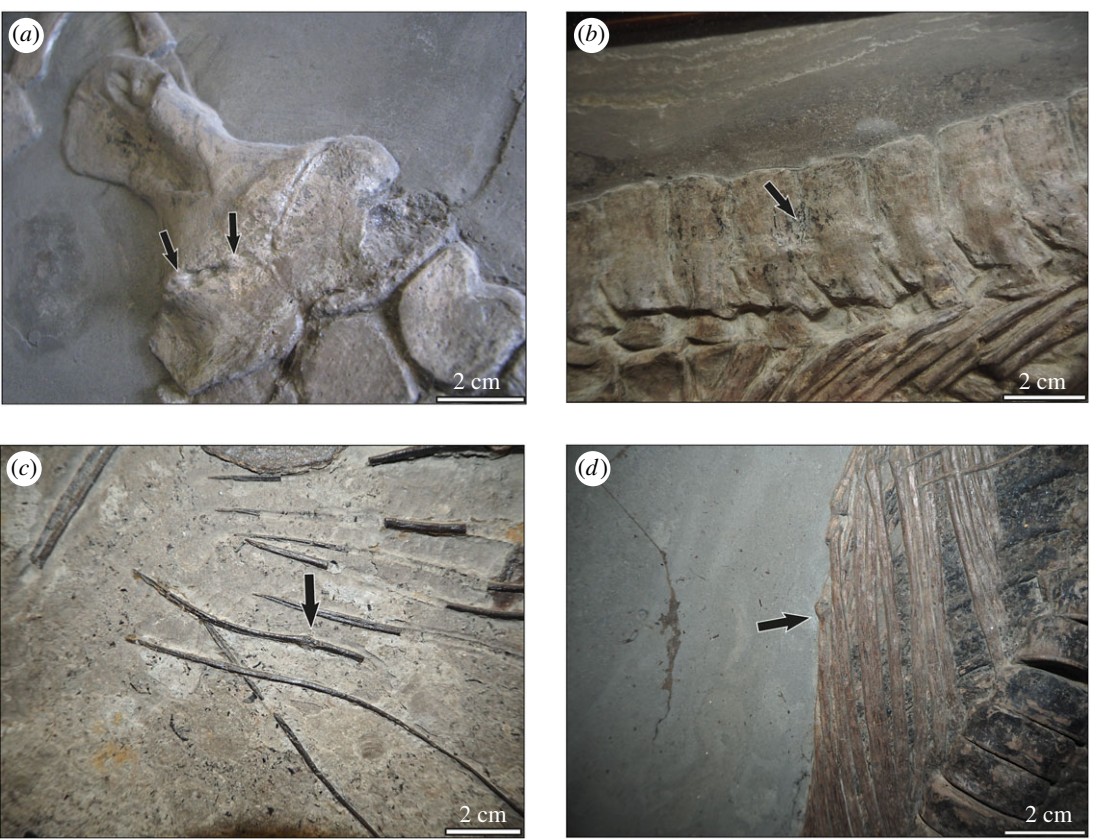

**Figure 1.** Examples of pathologies in ichthyosaurs from the Posidonienschiefer Formation. (*a*) GPIT no. 83. *Stenopterygius uniter*. Arrows indicate the ankylosed femur and fibula. (*b*) GPIT/RE/7301. *Stenopterygius quadriscissus*. The arrow indicates ankylosis between neural spines. (*c*) SMNS 81367. *Hauffipteryx typicus*. A fractured gastralium; the arrow indicates callus. (*d*) UMH dc7. *Stenopterygius* sp. A fractured rib; the arrow indicates callus.

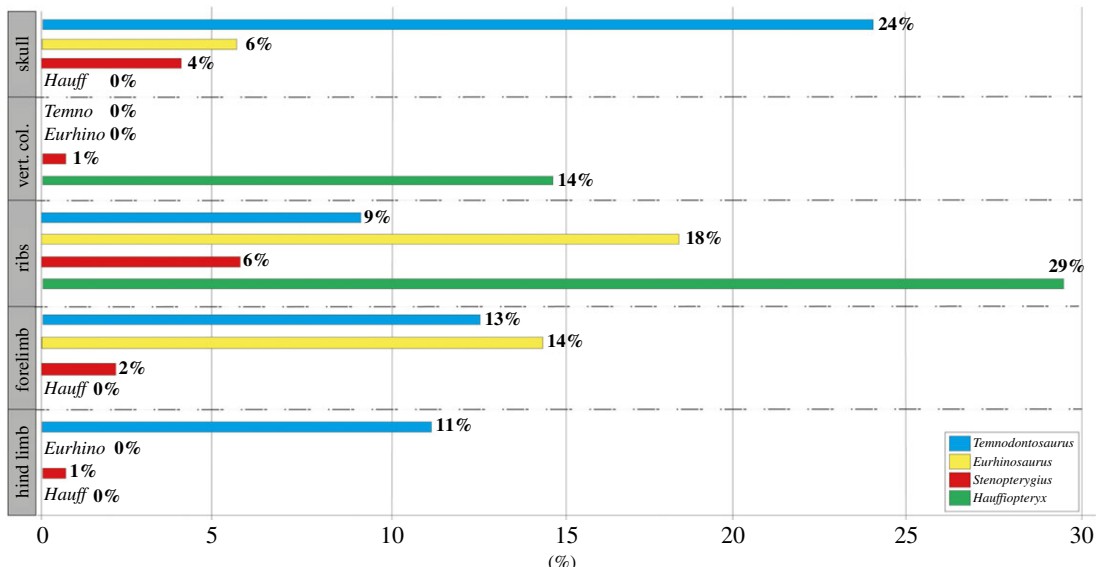

**Figure 2.** Prevalence of pathologies by taxon and anatomical region.

five specimens. Ankyloses in the phalanges of the forefin were observed in two specimens. No pathologies were observed in the vertebral column of *Temnodontosaurus*; pathologies in the pelvic girdle/hind fin (ankylosed phalanges) and ribs/gastralia regions (trauma with sign of healing) were observed in only one specimen each. The specimen with the pathological hindlimb also had ankyloses in the phalanges of the forefin [11].

## 3.2. Distribution of pathologies by anatomical unit

Based on all specimens and taxa, the most affected anatomical region was the ribs with 8% (13/174) showing pathologies, followed by the skull with 6% (12/203) and forelimb, in which 4% (7/186) were pathological. The least frequently pathological anatomical regions were the vertebral column and hind limb, with 2% (3/184) and 1% (2/154) of specimens affected.

## 3.3. Incidence of pathologies by taxon, size, ontogeny and stratigraphic level

### 3.3.1. Relative prevalence of pathologies by taxon

We analysed the relative prevalence of pathologies between the genera, with anatomical region as a covariate, omitting *Suevoleviathan* due to small available sample size. Both taxon and anatomical region explained significant amounts of variation in the prevalence of pathologies ($p < 0.05$) [12]; however, no anatomical region was significantly more or less pathological than the others ($p > 0.05$ for all). The interaction term (taxon × anatomical region) was not significant, and so was removed and the analysis rerun. *Stenopterygius* showed significantly (more than 2.9 times) fewer pathologies than all other taxa ($p = 0.04$).

### 3.3.2. Relative prevalence of pathologies by body length

Logistic regression with taxon size as the independent variable and anatomical region as a covariate indicated that large taxa were 2.4 times more likely to be pathological than small taxa ($p = 0.02$) [12], suggesting that adult body size is a significant factor influencing the incidence of pathologies in ichthyosaurs. However, because *Hauffiopteryx* is relatively rare, the large sample size of *Stenopterygius*, which overall has significantly fewer pathologies than any other genus, may be influencing this result. The interaction term (anatomical region × size) was omitted, as it did not explain substantially more of the variation in the sample.

### 3.3.3. Relative prevalence of pathologies through ontogeny

We tested the prevalence of pathologies in *Stenopterygius* across a range of size classes, a proxy for ontogenetic age, with anatomical region as a covariate. Ontogeny significantly affected the prevalence of pathologies [12], with the largest adults more than four times as likely to be pathological than juvenile specimens ($p = 0.008$). Small- and mid-sized adults did not exhibit significantly more frequent pathologies than juveniles (figure 3). Anatomical region was significantly informative as a covariate, but no one region was significantly more likely to be pathological than any other.

### 3.3.4. Relative frequency of pathologies over geological time

For this analysis, only *Stenopterygius* specimens with skeletal completeness greater than 50% were considered. Fourteen per cent (7/50) of pathological specimens were recovered in the beds underlying the carbon isotopic excursion, marking the onset of oceanic anoxia (beds $\varepsilon I_1$–$II_3$), 12.5% (3/24) were from the period of peak anoxia (beds $\varepsilon II_4$–$II_5$) and 6.5% (2/31) were from the overlying horizons $\varepsilon II_6$–III. Logistic regression did not recover significant differences in the frequency of pathologies between horizons [12], indicating that ecological change is not a major factor influencing the incidence of pathologies in this genus.

# 4. Discussion

Large numbers of previously unreported pathologies were observed in all taxa (figure 1) [12]. Systematic data collection indicates that there is much less disparity observed in the distribution of pathologies between anatomical regions than typically reported in the literature [10]. However, the low incidence of pathologies in the vertebral column in ichthyosaurs from the Posidonienschiefer Formation is consistent with other studies. This pattern is divergent from other large secondarily aquatic reptiles and mammals relying on axial propulsion, namely mosasaurid squamates and cetaceans, in which pathologies in the vertebral column are abundant [19–25]. Traumatic injuries dominate the observed pathologies in ichthyosaurs from the Posidonienschiefer Formation, consistent with [11] but again

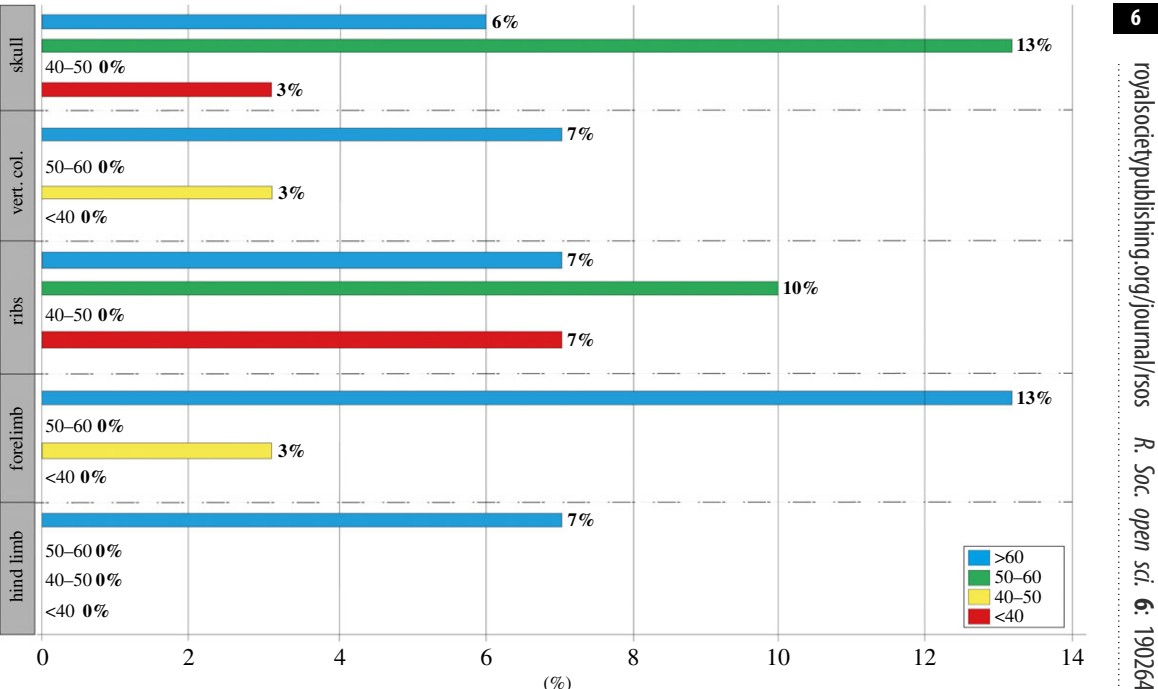

**Figure 3.** Prevalence of pathologies in *Stenopterygius* by size class, defined by mandibular length (cm).

differing from the broader literature [10]. Our failure to detect avascular necrosis in this survey is notable, as this pathology is prevalent in ichthyosaurs from the older Blue Lias and younger Oxford Clay Formations [26,27]. Its apparent absence in the Posidonienschiefer Formation may be attributed to the difficulty in identifying subsidence structures in strongly compressed slab-mounted specimens, or may reflect the absence of deep-diving behaviours in this ecosystem. Water depth in the Posidonienschiefer Formation is reconstructed as being slightly deeper than either the Blue Lias or Oxford Clay Formations (50–150 versus 10–50 m [16,28,29]); however, if water column hypoxia/euxinia was more widespread or of longer duration, prey may have remained concentrated near the surface.

The prevalence of skeletal injuries is known to differ between predators with distinct hunting strategies [2]. *Stenopterygius* exhibited significantly fewer osteopathologies than the other genera in our survey (contra [10]), indicating either a lower risk hunting strategy consistent with a diet of relatively small prey (e.g. [29]), or fewer aggressive interactions with conspecifics. However, the high prevalence of pathologies in the largest adults suggests a third possibility, namely that the *Stenopterygius* sample is on average ontogenetically younger than that of the other genera; in other words, had fewer years over which to accumulate traumatic damage to the skeleton. In support of the latter hypothesis, the prevalence of cranial pathologies in *Stenopterygius* increases sharply from 2% in juveniles and small adults to 10% in specimens with mandibular lengths greater than 50 cm (figure 3).

Hunting-related injuries are well documented among both marine and terrestrial vertebrate predators [30] and are predicted to be more frequent and severe among larger-bodied predators, which pursue proportionately larger prey [31]. However, non-fatal predator-induced injuries are also common and their incidence is affected by a range of factors, including predation rate, predator inefficiency, age structure of the prey population and survivorship of injured individuals [32–35]. Injuries incurred during intraspecific conflict are also likely: for instance, skeletal injuries to the ribs [20,36] and jaws [36] of odontocete cetaceans are attributed to aggressive interactions with conspecifics. Mid-level consumers [37] (*Stenopterygius*, *Hauffiopteryx*) differed from the larger taxa (*Eurhinosaurus*, *Suevoleviathan*, *Temnodontosaurus*) in the number, but not skeletal distribution, of observed pathologies, with traumas being the most commonly observed pathologies in both size classes. Consistent with terrestrial taxa [36,38], large body size does not appear to confer protection from skeletal trauma in ichthyosaurs, and, in fact, the reverse appears probable, suggesting that failed predation attempts account for a relatively low proportion of total observed injuries.

In extant vertebrates, the observed prevalence of skeletal pathologies increases with ontogenetic age ([2] and references therein; [35,39]). Consistent with the literature for other vertebrates [11,40], in *Stenopterygius*, more pathologies were observed in the largest-bodied adults than in juveniles, small

and mid-sized adults (figure 3). There are several potential underlying causes for this distribution. Osteologically immature individuals may have experienced more rapid healing, resulting in a smaller number of macroscopically visible pathologies [41,42]. Alternatively, small individuals may have experienced reduced survivorship following injury. However, large adults may show more skeletal pathologies than smaller individuals for another reason: whereas very large adults do not show substantially more frequent traumatic injuries than other size classes, they do exhibit an increased prevalence of articular disease and ankylosis in the forelimb, hind limb and vertebral column. These types of osteopathologies are typically associated with ageing.

The Posidonienschiefer Formation spans the early T-OAE, a minor mass extinction affecting primarily invertebrates [7], but also influencing ichthyosaur diversity and inter- and intraspecific body size [8]. We found no significant differences in the prevalence of skeletal pathologies in *Stenopterygius* immediately before, at the peak of and during the recovery phase following the anoxic event. The prevalence of pathologies is slightly, but not significantly, reduced (6.5 versus 12.5%) towards the top of the section [12]. This suggests that in this case, environmental stress is not related either to more frequent illness or injury (expected increase in the prevalence of pathologies) or to reduced survivorship following illness or injury (expected decrease in the prevalence of pathologies).

# 5. Conclusion

The appeal of palaeopathology is in its ability to create a dramatic picture of prehistoric animals interacting millions of years ago. However, the risk of over-interpreting isolated data points as providing information on usual behaviour for a taxon remains high. Most previous studies in marine reptiles, with the exception of those examining the prevalence of avascular necrosis (e.g. [27,43,44]), have been generally anecdotal in nature, reporting only the description of a specific pathology in one or two specimens (e.g. [45]) or suite of pathologies in a single specimen (e.g. [46]). Our study provides the first detailed and systematic analysis of the prevalence of pathologies in a Mesozoic marine system, and provides comparative baseline data for the prevalence of pathologies in ichthyosaurs and, more generally, Mesozoic marine reptiles occupying mid- to high trophic levels.

Of the palaeoecological variables analysed (genus, adult body size, ontogeny and environmental stress), genus, adult body size and ontogeny significantly affected the number of pathologies detected. Thus, ichthyosaurs appear to be similar to extant vertebrates in which pathologies accumulate in the oldest/largest members of a population, and larger taxa experience proportionately more frequent skeletal traumas. Extending this baseline survey across taxa and geological time periods will allow broader evolutionary and ecological concepts to be explicitly tested in a palaeoepidemiological framework.

Data accessibility. Data are available from the Dryad Digital Repository at: https://doi.org/10.5061/dryad.qm52585 [12].
Authors' contributions. J.M.P.-P. and E.E.M. designed the study and interpreted the results. J.M.P.-P collected and analysed the data. J.M.P.-P., E.E.M. and B.K. wrote the manuscript. All authors gave final approval for publication.
Competing interests. The authors declare no competing interests.
Funding. This research was funded by the Deutsche Forschungsgemeinschaft (DFG) Project MA 4693/4-1.
Acknowledgements. We thank R. Hauff (UMH), I. Werneburg (GPIT) and A. Schmid-Röhl (FWD) for collections access. We are also grateful to Davide Foffa and a second anonymous reviewer for helping us to improve our article with their observations.

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
