## [Reviewer comments · Royal Society Open Science]

Review History

RSOS-190264.R0 (Original submission)

Review form: Reviewer 1 (Davide Foffa)

Is the manuscript scientifically sound in its present form?

Yes

Are the interpretations and conclusions justified by the results?

Yes

Is the language acceptable?

Yes

Is it clear how to access all supporting data?

Yes

Do you have any ethical concerns with this paper?

No

Have you any concerns about statistical analyses in this paper?

No

Recommendation?

Accept with minor revision (please list in comments)

Comments to the Author(s)

See attached pdfs (Appendices A & B).

One (Review_DF) contains my review letter with attached general comments, and the second one other contain specifics comments embedded in the manuscript PDF.

Review form: Reviewer 2 (Bruce Rothschild)

Is the manuscript scientifically sound in its present form?

Yes

Are the interpretations and conclusions justified by the results?

Yes

Is the language acceptable?

Yes

Is it clear how to access all supporting data?

No

Do you have any ethical concerns with this paper?

No

Have you any concerns about statistical analyses in this paper?

Yes

Recommendation?

Major revision is needed (please make suggestions in comments)

Comments to the Author(s)

This is an interesting approach, perhaps limited by sample size. It should be emphasized that comparisons with prior publication prevalence is complicated by the search image utilized. So, absence of report of a certain type of pathology may simply represent its lack of interest (related to the study subject).

All non-fracture pathology (including abscesses/infections, be they isolated or associated with fractures) need to be illustrated and the few images are slightly out of focus or at least of inadequate resolution.

This is especially important, as articular ankyloses can be infectious or part of an underlying arthritis.

There is much self-citation, but the literature is more extensive and should be cited, especially

from primary references. There is more baseline data than is referenced. They cite Rothschild et al. (2012)

A common error that we all are guilty of, myself included, is to use the term "frequency." Frequency relates to occurrence in a given time interval. The correct term is "prevalence."

When statistical analysis is performed, the test should be revealed. Given the small numbers, I suspect that Fisher exact test would be required. Chi square is not valid if 25% or more of "boxes" have values of 5 or less. Beta (Type II error also needs to be reported) when no statistical difference is recognized.

Decision letter (RSOS-190264.R0)

11-Jun-2019

Dear Dr Pardo-Pérez

On behalf of the Editors, I am pleased to inform you that your Manuscript RSOS-190264 entitled "Palaeoepidemiology in extinct vertebrate populations: factors influencing skeletal health in Jurassic marine reptiles" has been accepted for publication in Royal Society Open Science subject to minor revision in accordance with the referee suggestions. Please find the referees' comments at the end of this email.

The reviewers and handling editors have recommended publication, but also suggest some minor revisions to your manuscript. Therefore, I invite you to respond to the comments and revise your manuscript.

- Ethics statement

- Data accessibility

<http://datadryad.org/submit?journalID=RSOS&manu=RSOS-190264>

- **Competing interests**

- **Authors' contributions**

- **Acknowledgements**

- **Funding statement**

Because the schedule for publication is very tight, it is a condition of publication that you submit the revised version of your manuscript before 20-Jun-2019. Please note that the revision deadline will expire at 00.00am on this date. If you do not think you will be able to meet this date please let me know immediately.

1) Identifying all the changes that have been made (for instance, in coloured highlight, in bold text, or tracked changes);

on behalf of Professor Rachel Wood (Associate Editor) and Jon Blundy (Subject Editor)
openscience@royalsociety.org

Reviewer comments to Author:

Reviewer: 1

Comments to the Author(s)

See attached pdfs.

One (Review_DF) contains my review letter with attached general comments, and the second one other contain specifics comments embedded in the manuscript PDF.

Reviewer: 2

Comments to the Author(s)

This is an interesting approach, perhaps limited by sample size. It should be emphasized that comparisons with prior publication prevalence is complicated by the search image utilized. So, absence of report of a certain type of pathology may simply represent its lack of interest (related to the study subject).

All non-fracture pathology (including abscesses/infections, be they isolated or associated with fractures) need to be illustrated and the few images are slightly out of focus or at least of inadequate resolution.

This is especially important, as articular ankyloses can be infectious or part of an underlying arthritis.

There is much self-citation, but the literature is more extensive and should be cited, especially from primary references. There is more baseline data than is referenced. They cite Rothschild et al. (2012)

A common error that we all are guilty of, myself included, is to use the term "frequency." Frequency relates to occurrence in a given time interval. The correct term is "prevalence."

When statistical analysis is performed, the test should be revealed. Given the small numbers, I suspect that Fisher exact test would be required. Chi square is not valid if 25% or more of "boxes" have values of 5 or less. Beta (Type II error also needs to be reported) when no statistical difference is recognized.

Author's Response to Decision Letter for (RSOS-190264.R0)

See Appendix C.

Decision letter (RSOS-190264.R1)

04-Jul-2019

Dear Dr Pardo-Pérez,

I am pleased to inform you that your manuscript entitled "Palaeoepidemiology in extinct

vertebrate populations: factors influencing skeletal health in Jurassic marine reptiles" is now accepted for publication in Royal Society Open Science.

on behalf of Professor Rachel Wood (Associate Editor) and Jon Blundy (Subject Editor)
openscience@royalsociety.org

Follow Royal Society Publishing on Twitter: [@RSocPublishing](https://twitter.com/RSocPublishing)
Follow Royal Society Publishing on Facebook:
<https://www.facebook.com/RoyalSocietyPublishing.FanPage/>
Read Royal Society Publishing's blog: <https://blogs.royalsociety.org/publishing/>

Appendix A

Dear Editor and Authors,

The manuscript I was sent to review is a brief review and innovative analysis of the distribution of pathologies in marine reptiles. The authors review the occurrence of pathologies (subdivided in different “types”) in ichthyosaurs from the well-sampled Posidonia Shale and through documented environmental/faunal shifts.

I consider this a well-written and organised manuscript that would only require minor changes before being ready for publication.

I particularly appreciated the hypothesis-driven approach and the impressive number of specimens examined by the authors. These factors, combined with simple and well-executed analyses contribute to a welcomed addition to our knowledge of marine reptile ecosystems. However, in my opinion, the biggest merit of this study is that it provides a powerful methodology that can be extended to other case study (e.g. other well-sampled intervals/ecosystems). Finally, similar approaches can be adopted to test prey/predator inferences and “feeding guilds” hypothesis in complex ecosystems. I believe that the authors could add some remarks about these possibilities in the abstract and conclusion.

I would argue that some minor additional details and clarifications are still needed before full acceptance. The majority of the suggested changes can be found in the attached PDF, and I will use this letter to highlight broader and more general points.

Accordingly, while I found the text well written and organised but the authors could add the following modifications to make it even clearer:

- 1) First of all the authors could spend some more words (one paragraph or two) explaining in more detail the pathologies they are analysing and what could cause them. This information, while not vital, would help clarifying which traumas can be used as evidence of animal interactions/predation. Until reading the Methods it was not clear what type of pathologies the authors were referring to (e.g. only bite marks?). It may be also worth specifying earlier than the Discussion that not all pathologies necessarily indicate a predator/prey interaction.
- 2) Clarifying some specialist terms and metrics (e.g. interaction term) would greatly benefit the clarity of the text. These should be explained in the methods, perhaps with reference of previous studies that have adopted similar techniques in support.
- 3) As previously mentioned the authors could use more effectively the Abstract to respectively highlight the merits of this manuscript but also justify their chosen methods and factors such as considered interval, decision of focusing on one group only etc..
- 4) The authors should carefully double-check their in-text reference and figure formatting. Multiple styles are currently adopted, including some odd and unnecessary formatting for manuscript “(e.g., (40))”

Specifics on these individual points can be found in the attached PDF.

Figures

The figures are generally well designed, informative and consistent in style. I particularly appreciated the use of figure 1 to illustrate different types of pathologies. The figures are

effectively referenced through the text (although there is some inconsistency in the capitalisation of subfigures in the in-text references).

Here I suggest some minor modifications that would improve the readability of the figures:

- 1) The text in figure 2 and 3 is really hard to read. Particularly the % and explanation of the colour code. Additionally, it would be appropriate double-checking that the colours in the figures are colour-blind friendly. The majority of figure-making software offers an option to check whether the colour combination is appropriate. Alternatively, the addition of a pattern on the graph bars would be sufficient.
- 2) I would personally enjoy a figure illustrating the partition of the skeletons adopted by the authors. I realise this has been done in a previous manuscript, but I think the text would benefit from it. Consider this as a suggestion though – I leave it to the authors' judgment deciding whether this extra figure (that could easily be added at the bottom of figure 1) would be appropriate or not.

Given the minor nature of these modifications I do not think that the manuscript will require an extra round of review. I look forward to see a final version of this manuscript, and if appropriate for the journal policy I would be happy to be contacted privately by the authors (davidefoffa@gmail.com or d.foffa@nms.ac.uk) for discussion and further clarifications on my comments.

Best regards,

Davide Foffa

Appendix B**ROYAL SOCIETY
OPEN SCIENCE****Palaeoepidemiology in extinct vertebrate populations:
factors influencing skeletal health in Jurassic marine
reptiles**

Journal:	Royal Society Open Science
Manuscript ID	RSOS-190264
Article Type:	Research
Date Submitted by the Author:	20-Feb-2019
Complete List of Authors:	Pardo-Pérez, Judith; Staatliches Museum für Naturkunde Stuttgart, Paleontology; Universidad de Magallanes, Vicerrectoría de Investigación y Postgrado Kear, Benjamin; Uppsala University, Museum of Evolution Maxwell, Erin; Staatliches Museum für Naturkunde Stuttgart, Paläontologie
Subject:	Palaeontology < EARTH SCIENCES
Keywords:	Palaeopathology, Palaeontology, Ichthyosauria, Palaeoecology, Posidonienschiefer Formation, Marine reptiles
Subject Category:	Earth science

Palaeoepidemiology in extinct vertebrate populations: factors influencing skeletal health in Jurassic marine reptiles

Judith M. Pardo-Pérez^{1,2}, Benjamin Kear³ and Erin E. Maxwell¹

¹Staatliches Museum für Naturkunde Stuttgart, Germany.

²Dirección de Investigación y Postgrado. Universidad de Magallanes, Punta Arenas, Chile.

³Museum of Evolution, Uppsala University, Uppsala, Sweden.

*Corresponding author

E-mail: judith.pardo-perez@smns-bw.de

Abstract

Palaeoepidemiological studies related to palaeoecology are rare, especially among marine taxa. Although such studies have the potential to provide information regarding ecosystem-level characteristics by measuring individual health, baseline data are lacking. In order to assess factors underlying the prevalence of pathologies in fossil marine reptiles, we surveyed ichthyosaurs from the Posidonienschiefer Formation (Early Jurassic: Toarcian) of Southwestern Germany. We considered five variables as potentially influencing the frequency of pathologies: taxon, anatomical region, body size, ontogeny, and environmental change, as represented by the early Toarcian Oceanic Anoxic Event. We found that the incidence of pathologies is dependent on taxon, with the small-bodied genus *Stenopterygius* exhibiting fewer skeletal pathologies than all other taxa. Larger-bodied genera show a higher incidence of pathology overall than smaller-bodied genera. Within *Stenopterygius*, we detected more pathologies in large adults than in juveniles and small- mid-sized adults. Stratigraphic horizon did not influence the incidence of pathologies in *Stenopterygius*, implying that palaeoenvironmental changes did not have a major effect on metrics of skeletal health in this system. Quantification of the occurrence of pathologies within taxa and across guilds is critical to constructing more detailed hypotheses regarding changes in the frequency of skeletal injury and disease through Earth history.

Keywords: Palaeopathology, Palaeontology, Ichthyosauria; Palaeoecology; Posidonienschiefer Formation; Marine reptiles

1. Introduction

[revised manuscript text omitted]

34. Abel O. Vorzeitliche Lebensspuren. G. Fischer; 1935. 644 p.
35. Buikstra JE. Healed fractures in *Macaca mulatta*: age, sex, and symmetry. *Folia Primatol.*
1975;23:140–8.
36. Foster SA. Wound healing: a possible role of cleaning stations. *Copeia.* 1985;875–80.
37. Aalami OO, Nacamuli RP, Lenton KA, Cowan CM, Fang TD, Fong KD, et al. Applications of a
mouse model of calvarial healing: Differences in regenerative abilities of juveniles and adults.
*Plast Reconstr Surg.* 2004;114(3):713–20.
38. Rothschild B, Martin LD. Avascular necrosis: occurrence in diving cretaceous mosasaurs.
*Science.* 1987;236(4797):75–7.
39. Rothschild BM, Storrs GW. Decompression syndrome in plesiosaurs (Sauropterygia: Reptilia). *J*
*Vertebr Paleontol.* 2003;23(2):324–8.
40. Surmik D, Szczygielski T, Janiszewska K, Rothschild BM. Tuberculosis-like respiratory infection
in 245-million-year-old marine reptile suggested by bone pathologies. *R Soc Open Sci.*
2018;5:180225.
41. Sassoon J, Noè LF, Benton MJ. Cranial anatomy, taxonomic implications and palaeopathology
of an Upper Jurassic Pliosaur (Reptilia: Sauropterygia) from Westbury, Wiltshire, UK.
*Palaeontology.* 2012;55(4):743–73.

Figure captions

**Figure 1.** Examples of pathologies in ichthyosaurs from the Posidonienschiefer Formation **a)** GPIT no.
83. *Stenopterygius uniter*. Arrows indicate the ankylosed femur and fibula. **b)** GPIT/RE/7301.
*Stenopterygius quadriscissus*. Arrow indicates ankylosis between neural spines. **c)** SMNS 81367.
*Hauffipteryx typicus*. A fractured gastralium; arrow indicates callus. **d)** UMH dc7. *Stenopterygius* sp. a
fractured rib; arrow indicates callus.

**Figure 2.** Frequency of pathologies by taxon and anatomical region.

**Figure 3.** Frequency of pathologies in *Stenopterygius* by size class, defined by mandibular length (cm).

Supplementary Information.

**1. Raw data.** A specimen-level database of material examined during the course of this survey.
**2. Collated tables used in analyses and tabular results of logistic regression.**

Ethics. No ethical assessment was required prior to conducting our research.

Data accessibility. Our dataset has been uploaded to the Dryad Digital Repository
(<https://datadryad.org/review?doi=doi:10.5061/dryad.qm52585>)

Authors' contributions. J.P.P. and E.E.M. designed the study. J.P.P. collected and analyzed the data.
9 J.P.P. and E.E.M. interpreted the results. J.P.P., E.E.M. and B.K. wrote the manuscript. All authors gave
final approval for publication.

Competing interest. The authors declare no competing interests.

Funding. This research was funded by the Deutsche Forschungsgemeinschaft (DFG) Project MA
4693/4-1.

Acknowledgments. We are deeply grateful to R. Hauff (UMH), I. Werneburg (GPIT) and A. Schmid-Röhl
(FWD) for their kindness in allowing us access to the collections and to the Deutsche
Forschungsgemeinschaft (DFG) for funding our research.

Figure 1. Examples of pathologies in ichthyosaurs from the Posidonienschiefer Formation a) GPIT no. 83. *Stenopterygius uniter*. Arrows indicate the ankylosed femur and fibula. b) GPIT/RE/7301. *Stenopterygius quadricissus*. Arrow indicates ankylosis between neural spines. c) SMNS 81367. *Hauffipteryx typicus*. A fractured gastralium; arrow indicates callus. d) UMH dc7. *Stenopterygius* sp. a fractured rib; arrow indicates callus.

197x148mm (300 x 300 DPI)

Figure 2. Frequency of pathologies by taxon and anatomical region.

199x103mm (300 x 300 DPI)

Figure 3. Frequency of pathologies in *Stenopterygius* by size class, defined by mandibular length (cm).

191x113mm (300 x 300 DPI)

Appendix C

Reviewer's response

Reviewer 1

- 1) First of all the authors could spend some more words (one paragraph or two) explaining in more detail the pathologies they are analyzing and what could cause them. This information, while not vital, would help clarifying which traumas can be used as evidence of animal interactions/predation. Until reading the Methods it was not clear what type of pathologies the authors were referring to (e.g. only bite marks?). It may be also worth specifying earlier than the Discussion that not all pathologies necessarily indicate a predator/prey interaction.
Author: explanations to each suggested sections were included.
- 2) Clarifying some specialist terms and metrics (e.g. interaction term) would greatly benefit the clarity of the text. These should be explained in the methods, perhaps with reference of previous studies that have adopted similar techniques in support.
Author: explanation was included.
- 3) As previously mentioned the authors could use more effectively the Abstract to respectively highlight the merits of this manuscript but also justify their chosen methods and factors such as considered interval, decision of focusing on one group only etc..
Author: a more detailed explanation was included to the abstract
- 4) The authors should carefully double-check their in-text reference and figure formatting. Multiple styles are currently adopted, including some odd and unnecessary formatting for manuscript "(e.g., (40))"
Author: the mistakes were corrected.
- 5) Specifics on these individual points can be found in the attached PDF.
Author: We included the corrected version of the manuscript with track changes, following each of the suggested corrections of reviewers.
- 6) The text in figure 2 and 3 is really hard to read. Particularly the % and explanation of the colour code. Additionally, it would be appropriate double-checking that the colours in the figures are colour-blind friendly. The majority of figure-making software offers an option to check whether the colour combination is appropriate. Alternatively, the addition of a pattern on the graph bars would be sufficient.
Author: The size of the text in the figures was increased according to the reviewer observation. The colors used in the graphics are color-blind friendly.
- 7) I would personally enjoy a figure illustrating the partition of the skeletons adopted by the authors. I realise this has been done in a previous manuscript, but I think the text would benefit from it. Consider this as a suggestion though – I leave it to the authors' judgment deciding whether this extra figure (that could easily be added at the bottom of figure 1) would be appropriate or not.
Author: We appreciate the reviewer suggestion but this has figured previously by Beardmore and Furrer (2016) (mentioned within the manuscript) and our classification is described using fairly self-evident descriptive labels (e.g., 'skull', 'vertebral column' etc.), making a figure unnecessary.

Reviewer 2

- 1) All non-fracture pathology (including abscesses/infections, be they isolated or associated with fractures) need to be illustrated and the few images are slightly out of focus or at least of inadequate resolution.

Author: The images selected in Figure 1 are of the best resolution that we were able to get, given the poor light conditions in mounted skeleton and depositories. Most of the material is polished by mechanical preparation because they correspond to old collections prepared with old techniques, therefore the nature of ankyloses in the cases of Posidonia shale cannot be determinate. More detailed figures of each pathology was published in Pardo-Pérez et al., (2018b), this publication is cited in the text.

- 2) There is much self-citation, but the literature is more extensive and should be cited, especially from primary references. There is more baseline data than is referenced. They cite Rothschild et al. (2012)

Author: We included more references according to the reviewer observation.

- 3) A common error that we all are guilty of, myself included, is to use the term "frequency." Frequency relates to occurrence in a given time interval. The correct term is "prevalence."

Author: We corrected the mistake according to the reviewer observation.

- 4) When statistical analysis is performed, the test should be revealed. Given the small numbers, I suspect that Fisher exact test would be required. Chi square is not valid if 25% or more of "boxes" have values of 5 or less. Beta (Type II error also needs to be reported) when no statistical difference is recognized.

Author: The complete results of the statistical analysis was included.